# Loss of Continuity of Care in Pediatric Neurology Services during COVID-19 Lockdown: An Additional Stressor for Parents

**DOI:** 10.3390/children9060867

**Published:** 2022-06-10

**Authors:** Serena Cesario, Consuelo Basile, Matteo Trevisan, Federica Gigliotti, Filippo Manti, Rita Maria Esposito, Giuseppe Abbracciavento, Mario Mastrangelo

**Affiliations:** 1Division of Child and Adolescent Neurology and Psychiatry, Department of Human Neuroscience, Sapienza University of Rome, Via dei Sabelli 108, 00185 Rome, Italy; serena.cesario@uniroma1.it (S.C.); consuelo.basile@uniroma1.it (C.B.); federica.gig@uniroma1.it (F.G.); filippo.manti@uniroma1.it (F.M.); 2Institute for Maternal and Child Health, IRCCS “Burlo Garofolo”, Via dell’Istria 65, 34137 Trieste, Italy; matteo.trevisan91@gmail.com (M.T.); giuseppe.abbracciavento@burlo.trieste.it (G.A.); 3Department of Medicine, Surgery and Health Sciences, University of Trieste, Piazzale Europa 1, 34127 Trieste, Italy; 4Department of Psychology, Sapienza University of Rome, Via dei Marsi 78, 00185 Rome, Italy; ritamaria.esposito@uniroma1.it; 5IRCSS Foundation Santa Lucia, Via Ardeatina 306/354, 00179 Rome, Italy

**Keywords:** COVID-19 pandemic, caregiver, perceived stress, children

## Abstract

**Background**. This study aimed to investigate the consequence of the COVID 19-related lockdown on the well-being of children with neurological and neurodevelopmental disorders and the repercussion on parental stress during the period 9 March 2020–3 May 2020. **Methods**. A web-based survey was shared via mail with the parents of children affected by chronic neurologic disorders and neurodevelopmental disorders in the continuity of care in two Italian tertiary centers, independently by the severity of the diseases and the required frequency of controls. For each patient, they were asked to identify a single main caregiver, among the two parents, to fill in the questionnaire. Parental stress was measured via the Perceived Stress Scale (PSS). Statistical analysis was performed with IBM SPSS Statistics version 25. The differences between the clinical groups were performed with one way ANOVA. The dimensional effect of the clinical variables on outcome was evaluated by multiple linear regression analysis. **Results**. The survey was completed by 250 parents (response rate = 48.9 %). Sars-Cov2 infection was reported in two patients only. A total of 44.2% of the patients had completely interrupted school activities while 70% of parents underwent changes in their job modalities. Health care services were disrupted in 77% of patients. Higher PSS scores were detected in the parents of children with neurodevelopmental disorders (*p* = 0.035). **Conclusions**. The loss of continuity of care during the lockdown must be considered as a risk factor for parents caring for children with chronic neurologic diseases and neurodevelopmental disorders in further phases of the current pandemic.

## 1. Introduction

The Italian Government introduced severe restrictive measures (“lockdown”) against COVID-19 on the entire national territory between 9 March 2020 and 3 May 2020 [1,2,3,4]. These measures slowed the diffusion of the viral infection, but resulted in relevant social and economic costs in several sectors including the medical assistance of non-COVID-19 diseases [4]. Many medical visits, diagnostic workups, and follow-up evaluations were suspended because all of the resources of the Health National System were addressed toward the management of the COVID-19 emergency [4].

The direct consequences of COVID-19 on pediatric patients suffering from chronic neurological or neurodevelopmental diseases involved a minor number of patients while the psychological impacts resulting from the social limitations and damage resulting from the reduction in medical and rehabilitative follow-up were probably higher [5,6,7]. Comparable negative consequences on the children’s well-being were also caused by the increased risk of experiencing stress and negative emotions in the parents [8]. Table 1 summarizes the few available studies including five cross-sectional surveys on the social and health care nodes and the caregivers’ stress in the field of pediatric neurology since the onset of the pandemic.

The present study aimed to analyze the impact of COVID 19-related lockdown on the parental perceived stress and, subsequently, on the health and the care of children with chronic neurological and neurodevelopmental diseases through a parental self-reporting survey about the consequences of the interruption of in-presence school, medical periodical follow-up, and rehabilitation services. These descriptors were judged to represent the main negative social and clinical changes in the daily life of families of children with neurological diseases.

## 2. Patients and Methods

The parents of patients with chronic neurological or neurodevelopmental disorders followed at the Unit of Child Neurology and Psychiatry of “Azienda Ospedaliera Universitaria Umberto I Hospital-Sapienza Università di Roma” in Rome (Italy), the Unit of Child Neurology and Psychiatry and the Unit of Rare Diseases of the “Burlo Garofalo” pediatric Hospital in Trieste (Italy) were considered as eligible for the study. 

Participation in the study was voluntary and no financial compensation was given to the recruited participants. A specific online-consent form for the collection and the publication of the data was signed before the enrollment according to the Italian legislation. The study obtained consent from the Institutional Ethic Committees (Institutional Review Border of IRCSS “Burlo Garofolo Pediatrc Hospital”, IRB code 10/20, approved on October 2020).

The list of consecutive recoveries or follow-up visits that were realized in the two centers between 4 May 2020 and 31 May 2020 was used as a source for recruiting the potential participants.

The inclusion criteria included: (a) the patient had to be in the continuity of care without discriminations based on the severity of the diseases and the required frequency of the controls; and (b) for each patient, they were asked to identify a single main caregiver, among the two parents, to fill in the questionnaire. 

The survey was shared via email after an explanatory phone call. The collection of questionnaires lasted from 1 June to 30 July 2020 with a normal distribution of the completed forms during the whole period. 

Participants were asked to fill in an anonymous online questionnaire in two parts (available at: https://forms.gle/6jbDYNDhrHPFDiS7A (accessed on 15 April 2022); see an English version in Appendix A) focused on the lockdown period (9 March 2020–3 May 2020): (a) the first part included general information about the children’s diseases and their trend during the lockdown but also the families’ habits and their changes, especially in the care and the needs of the children in this period; and (b) the second part included the Perceived Stress Scale (PSS), in order to measure the degree to which lockdown was appraised as stressful (see Appendix A) [9]. The first part included multiple choice answers including the possibility of an additional open answer (see Appendix A). No missing data were possible because the survey could not be uploaded if the participants did not fill in all the sections. 

The survey covered all the possible lockdown-related stressors that were correlated with the management of the health care needs of the patients and required less than 5 min to be completed.

Three groups were defined according to the patient’s diagnosis: (a) epilepsy of unknown etiology; (b) neurodevelopmental disorders; and (c) genetic-metabolic diseases. A predominant symptom, according to clinical judgement, was identified for the assignment of patients with comorbidities (e.g., epilepsy and neurodevelopmental disorders) to one of the above-mentioned groups. Patients presenting with epilepsies with a known genetic etiology were assigned to the third group.

All statistical analyses were conducted using IBM SPSS Statistics version 25.0 (SPSS Inc., Chicago, IL, USA). A *p* value < 0.05 represented the statistical significance for all tests. Data were reported as frequencies and percentages for discrete variables as well as the means and standard deviations for continuous variables. The chi-squared test (χ2) was used to reveal the differences between and within groups in the discrete variables investigated, while one-way analysis of variance (ANOVA) was carried out to test the between and within group differences in continuous variables. Bonferroni corrections were applied to reduce problems associated with multiple comparisons. The dimensional effect of clinical variables on the outcome was evaluated by multiple linear regression analysis.

## 3. Results

### 3.1. Demographic and Clinical Data

The survey was sent to 511 families, and it was filled in by the parents of 250 patients (response rate = 48.9%; the optimal sample size according to https://www.surveysystem.com/sscalc.htm (accessed on 15 April 2022) was 152 participants). The responders were mainly the mothers (71.5%). The cohort of the analyzed patients included: 96 patients with epilepsies of unknown etiology (38%), 88 patients with neurodevelopmental disorders (35%), 66 patients with genetics-metabolic diseases (33%; this group included 22 patients with genetically determined epilepsies). The mean age of the patients was 8 years and 9 months (age range: 12 months–37 years) and 53% of them was male. The age distribution among the three above-mentioned groups is indicated in Figure 1.

Before lockdown, 238 patients (95.2%) attended school while 154 patients (61.6%) took advantage of regular rehabilitation/occupational therapy

Two patients (0.8%) tested positive for COVID-19. The disease presented with gastrointestinal symptoms in one patient and with an asymptomatic course in the other. A total of 6% of the participants reported at least one familial case of COVID-19 infection. 

### 3.2. The Impact of Lockdown on Children Health and Care According to Parental Perspective

School closure resulted in a complete interruption of any teaching experience in 114 patients (44.2%), who did not have access to remote lessons. Remote teaching was positively judged by almost all of the parents with children involved in this method (71%) because of the chance of a continuing relationship with their mates and teachers and the maintenance of some sort of regularity. Despite the positive impression, the parents reported some critical issues such as unavailable or stable Internet connection, standard technologies non-suitable for disabilities, and “virtual” relationships with peers insufficient for the children’s well-being.

Lockdown changed the parents ‘work activities in 70% of the cases via layoffs (27%), working from home (39%), or reduction/suspension (38%).

Health and welfare services were suspended for 77% of patients. According to the parents, the children were negatively affected by the lack of rehabilitation services (67%) and the loss of the usual clinical and instrumental checks (45%). A total of 79.1% of parents were able to contact doctors, mostly by phone call or email. Thirty-seven patients (15%) needed urgent specialist examination during the lockdown period and in 68% of cases for reason of their neurological condition: eight patients underwent hospitalization, four patients accessed the emergency room, 17 patients needed an urgent outpatient visit with their neurologist, and the remaining patients were managed at home by their pediatrician.

Access to remote rehabilitation services was available for 58 patients (22.5%). Only 5% of parents found this method to be adequate for the therapeutic goals; nevertheless, most of them provided positive feedback on telerehabilitation because it allowed for a continuing relationship between the children and therapists (48%) and provided support to the parents, involving them in rehabilitation activities (38%). 

During the lockdown, the clinical conditions remained stable for most patients (83%) with a worsening in 15% (40 patients) of cases. Behavioral deteriorations were observed in 35% (86 patients) of cases. 

### 3.3. The Impact of Lockdown on Parental Distress

Perceived stress, as measured by PSS, was high in 37 parents (14%); moderate in 146 parents (58%); and low in 70 (28%). No significant correlation was found between the PSS score and age or gender of the patients (*p* = 0.06 and *p* = 0.48, respectively). The parents of children affected by neurodevelopmental disorders had higher levels of stress than other parents (F_(2;247)_ = 3.391, *p* = 0.035; Table 2).

Higher scores on the PSS were obtained by parents who underwent job changes (*p* < 0.05) while no significant perceived effects were associated with the presence of remote homeschooling and remote rehabilitation services (*p* value of 0. 43 and 0.58, respectively; Table 3).

Another significant stressor was represented by the loss/discontinuation of specialist follow-up programs and the need for urgent medical care, while no relevant effects were associated with the inability to contact the referring specialist (Table 3 and Table 4).

The three analyzed diagnostic groups did not differ in terms of the perceived clinical worsening by parents (chi-square = 7.71; *p* = 0.26). A significant behavioral deterioration (chi-square = 16.12; *p* < 0.05) was detected in all three groups, independent of the etiologies, with a higher negative perceived impact in patients with a neurodevelopmental disorder (45% of patients versus 29% of patients with epilepsy of unknown etiology and 30% of patients with genetic-metabolic diseases). A marginally significant perceived worsening in the functioning and adaptive behavior was assessed in all three diagnostic groups (chi-square = 16.66; *p* = 0.05). 

## 4. Discussion

This study confirmed that the COVID-19 pandemic had broader effects than infections only on several aspects of daily life and health care. The survey highlighted a negative impact of lockdown-related limitations and closures on the parental perception of the clinical conditions and the adequacy of access to health care services for children with chronic neurological and neurodevelopmental disorders. This negative perception contributed to an increase in the number of parental stressors and probably affected the patients’ care and quality of life. The most significant difficulties were reported by the parents of children with neurodevelopmental disorders because of a higher frequency of behavioral disturbances. This finding is consistent with previously published data that had shown more remarkable consequences of restrictions in children with autism spectrum disorder (ASD) and attention deficit hyperactivity disorder (ADHD) than in pediatric patients with other chronic disorders [9]. The higher vulnerability of these patients resulted from the known negative impact of changes in their daily routine and the limitations in the preferred activities in children with an autism spectrum disorder [10]. Data from a U.S. survey suggested that higher levels of stress in the parents of autistic children were mainly related to isolation, fear of illness or clinical worsening because of the lack of specialist care, economic worries, the severity of autistic symptoms, and younger age [11,12]. An increase in behavioral disruption was also reported in a significant proportion of patients with other diseases, ranging between 30–35.9% of children with epilepsy and 55% of children with cerebral palsy, neuromuscular disorders, and other neurological diseases [13,14,15]. The lack of human resources in providing health care and psychological support were perceived as the main stressors by parents in these last cases [15]. A few other studies have shown several additional behavioral changes in children with special needs during isolation and quarantine [7,13,16,17,18,19]. In these contexts, children might become more demanding with their parents and experience feelings of impatience, annoyance, and hostility, in addition to the fear of infection and frustration for the loss of their daily routine [7,13,16,17,18,19]. These reactions might complicate the global support granted by caregivers and should be considered in designing specific pathways for psychological support for parents [7,13,16,17,18,19].

The level of parental stress during the Sars-COV2 pandemic was assessed as significantly higher in the parents of children with chronic physical and/or chronic mental conditions than in the parents of healthy children [16]. The main stressors that were identified by the parents of children with chronic neurological diseases included: perception or fear of clinical worsening or increased clinical risk for their children, loss of regular health checks, inability to reach the referring neurologist using telemedicine resources, loss of economic income, anxiety and/or depression of the caregivers, and difficulties in finding specific drugs on the market (e.g., antiepileptic drugs) [13,18,19]. The interruption of clinical follow-up and rehabilitative programs represented the most relevant stressor for parents, in the herein reported, cohort, even if the experience with remote technologies was not completely negative and most of the caregivers had no difficulties in contacting the referring specialists. Remote schooling, telemedicine, and telerehabilitation services were perceived as partially useful to replace the direct relationship of children with medical specialists, therapists, support teachers, and mates while the suspension of medical care was considered more stressful than the closure of schools and rehabilitation services. The relevance of these stressors was independent of the etiology of the diseases of the reported children. This aspect suggests that parental worries and fears might be far more correlated with the loss of certainty about follow-up planning than with the basic clinical conditions. The relationship between the mentioned stressors and the fears of eventual clinical worsening might be a subject for debate, but both our data and the ones obtained in previously published cohorts suggested that this perception might be high in the caregivers [13,14]. These fears seem to be correlated with an effective increase in seizure frequency, which ranged between 13.3 and 14%, in two different cohorts of children with epilepsy in Spain and Italy, respectively [13,14]. 

The main limitations of our study include the cross-sectional design, the bias selection given by the voluntary participation of parents, the small size of the sample that might underrepresent the effective population of patients and parents referring to the two involved tertiary centers, the self-reported nature of all the collected data, and the possible effect of confounders. Another possible confounding effect was given by the different diffusion of the infections in the regions of the two enrolling centers, even if these two hospitals have a national catchment area. A further source of bias selection might be represented by the selection of patients who referred to two specialistic tertiary centers with a possible exclusion of children suffering from less severe neurological diseases. Moreover, the feelings and other events experienced by parents during the lockdown were recorded in the aftermath using a self-reporting questionnaire and no previous data on parental distress before and after the pandemic were available to compare the differences (e.g., behavioral changes were perceived as lockdown correlated by parents, but they might also be influenced by other individual and social factors). Similarly, the specific burden for neurological and neurodevelopmental disorders of the reported results was not quantifiable because no comparable data were available on children with non-neurological disorders.

## 5. Conclusions

Social distancing limited the Sars-CoV2 outbreak during the so-called “first wave” and caused a reorganization of health care settings and clinical practice that have affected the well-being of families caring for pediatric neurologic children. It seems necessary to adapt the health response to the families’ needs since the continuity of care for chronic patients could be considered as a protective factor by their parents and related to the caregivers’ mental health. Higher levels of stress in this at-risk population seems to be related to the suspension of the usual clinical checks. The maintenance of high-quality medical assistance for patients with diseases other than COVID-19 will represent a remarkable challenge for all health care systems during the next phases of the current Sars-CoV2 pandemic.

## Figures and Tables

**Figure 1 children-09-00867-f001:**
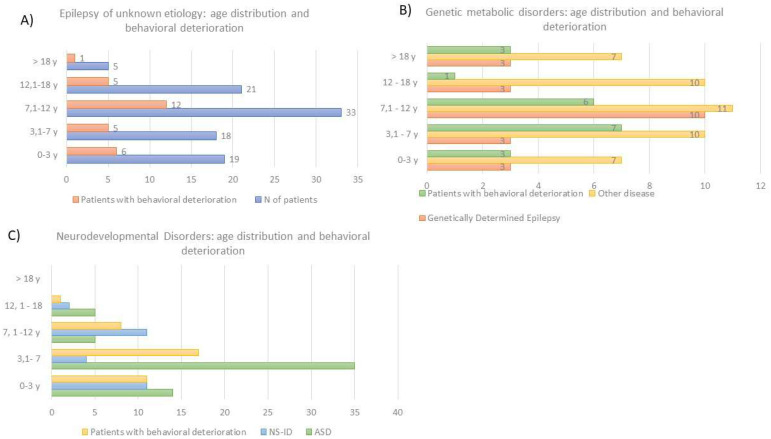
The age distribution among the three groups of diagnosis that were analyzed in this study: (**A**) Patients with epilepsy of unknown etiology; (**B**) Patients with genetic-metabolic diseases (including patients with genetically determined epilepsy); (**C**) Patients with neurodevelopmental disorders (including patients with autism spectrum disorder, patients with developmental delay, and patients with non-syndromic intellectual disability). For each group, the distribution of patients with behavioral deterioration is also reported. LEGEND: y = years; ASD = autism spectrum disorder; NS-ID = non syndromic intellectual disability.

**Table 1 children-09-00867-t001:** The main studies about the health care nodes and caregiver’s stress in children with neurological and neurodevelopmental disorders during the COVID-19 pandemic.

Reference	Country	Study Type	Number of Participants	Age-Range of Patients	Diseases	Main Findings of the Study	Main Limitations
Manning et al., 2020	Michigan, USA	Cross-sectional survey	471 (459 caregivers, 12 self-responders)	2–46 years (M 11.8 SD 7.9)	Autism	- Higher levels of stress in caregivers correlated with younger age, greater severity of symptoms and greater intensity of services pre-COVID-19 - Major stressors: therapeutic service disruption, finances, and illness	- Self-selection bias - Response rate not computable
Grumi et al., 2020	Italy	Cross-sectional survey	84 caregivers	1–15 years (M 6.13; SD 3.44)	- Autism - Psychomotor delay - Cerebral palsy, - Emotional-behavioral problems - Severe prematurity - Genetic or metabolic Syndromes - Epilepsy - Neuromuscular diseases - Primary sensory impairment	- Highest sources of psychological distress in caregivers were worries for the contagion and concerns about child development without proper services support	- Small sample size - Psychometric features of the survey not tested - No relevant descriptors for the reported cohort
Aledo-Serrano et al., 2020	Spain	Cross- sectional survey	277 caregivers	NA (M 12.4 years)	- Genetic developmental and epileptic encephalopathies (DEE)	- Increase of seizure frequency and behavioral deterioration during lockdown in patients with genetic DEE - Seizure increase was associated with age and difficulties in finding drugs on the market - Behavioral deterioration correlated to type of epilepsy, living in a home without a terrace or yard, and caregivers’ anxiety	- Cross sectional design
Trivisano et al., 2020	Italy	Cross-sectional survey	3321 (3209 caregivers, 112 self-responders)	NA 0–1 year = 72 patients; 2–5 years = 529 patients; 6–12 years = 1394 patients; 13–18 years = 746 patients; >18 years = 580 patients	- Epilepsy	- Greater effect of pandemics on comorbidities, mostly behavioral and sleep disturbances, than on seizures - Usefulness and convenience of telemedicine (avoiding long journeys and saving money)	- Response bias
Cacioppo et al., 2020	France	Cross-sectional survey	1000 caregivers (829 were parents of children with neurological diseases)	1–18 years (M 9.5 SD 4.8)	- Cerebral palsy - Genetic diseases or congenital malformations - Neuromuscular - Other neurological lesions	- Lockdown had negative effects on morale, behavior, and social interactions - Main parental concern was rehabilitation, and their main difficulty was mental load and lack of help and support	- Response bias - Representativeness of responders not ensured

Legend: M = mean; SD = standard deviation; NA = not available; DEE = developmental and epileptic encephalopathies.

**Table 2 children-09-00867-t002:** Multiple comparisons of the Perceived Stress Scale mean scores (post-hoc test).

Groups of Diagnosis		Difference of Mean	SE	95% CI	*p*
Epilepsy of unknown etiology					
	Neurodevelopmental disorders	−2.399	1.023	−4.41; −0.38	0.020 *
	Genetic-metabolic disease	0.752	1.098	−1.41; 2.91	0.494
Genetic-metabolic diseases					
	Epilepsy of unknow etiology	−0.752	1.098	−2.91; 1.41	0.497
	Neurodevelopmental disorders	−3.151	1.127	−5.37; −0.93	0.006 **
Neurodevelopmental disorders					
	Epilepsy of unknow etiology	2.399	1.023	0.38; 4.41	0.020 *
	Genetic-metabolic diseases	3.151	1.127	0.93; 5.37	0.006 **

LEGEND: SE= standard error of B; 95% CI = 95% confidence interval, * *p* < 0.05; ** *p* < 0.01.

**Table 3 children-09-00867-t003:** A comparison of the Perceived Stress Scale mean score among the parents related to the presence of health care, remote schooling, and remote rehabilitation during the lockdown.

Access to:	Preserved Access		Lacking Access		*p* Value
	Mean PSS Score (M + SD)	No. patients/ participants	Mean PSS Score (M + SD)	No. patients/ participants	
Health care	15.65 ± 6.2	54/250	17.78 ± 7.1	196/250	0.043 *
School	16.99 ± 6.7	136/250	17.69 ± 7.3	114/250	0.43
Rehabilitation	17.75 ± 7.1	58/250	17.7 ±6.9	192/250	0.58

LEGEND: M = mean; SD = Standard deviation; * *p* < 0.05.

**Table 4 children-09-00867-t004:** The multiple regression linear model examining the association between the PSS total score and potential stressors.

Stressors	R	F	B	SE	95% CI
	0.242 *	3.086 *			
Job changes			2.367 *	0.941	0.515–4.220
Loss of continuity of care			2.269 *	1.067	0.167–4.370
Need of emergency care			2.574 *	1.266	0.080–5.067
Telerehabilitation			0.358	1.058	−1.725–2.442
Online homeschooling			−0.277	0.896	−2.041–1.487

Legend: F = F value of the total model, B = unstandardized regression weight, SE= standard error of B; 95% CI = 95% confidence interval of B; * *p* < 0.05.

## Data Availability

Data are available on request from the authors.

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
