# Peer review of "Loss of Continuity of Care in Pediatric Neurology Services during COVID-19 Lockdown: An Additional Stressor for Parents"

_children, 2022, doi:10.3390/children9060867_

Round 1
Reviewer 1 Report
Comments to the authors
The paper is well written and some minor improvements are needed.
Major comments
Abstract
More methodological details should be included (e.g. statistics, subjects completing the survey).
Introduction
The Authors should better clarify that the study explore parents and not children perceived stress.
Methods
- More details about the PSS should be interesting for the readers.
- Statistical analysis should be improved and overall better clarified in methods section.
Results
- It is not clear what is reported in table 3.
- Multiple comparisons correction should be performed.
Author Response
The paper is well written, and some minor improvements are needed.
We would acknowledge the reviewer for his positive judgement and for his valuable suggestions.
Major comments
Abstract
More methodological details should be included (e.g. statistics, subjects completing the survey).
We added the required methodological info (page 1, lines 22-27)
Introduction
The Authors should better clarify that the study explore parents and not children perceived stress.
We better clarified that “the present study aimed to analyze the impact of COVID 19-related lockdown on the parental perceived stress …. (page 3, lines 59-60)”.
Methods
More details about the PSS should be interesting for the readers.
Statistical analysis should be improved and overall better clarified in methods section.
We added more details about PSS version and statistical methods (page 4; lines 108-116)
Results
It is not clear what is reported in table 3.
Multiple comparisons correction should be performed.
We rewrote the legend of the new Table 2 (according to the request of the other reviewer we put the previous Table 3 before the previous Table 2) that became “Multiple comparisons Perceived Stress Scale mean scores (post-hoc test). LEGEND: *p < .05; **p<.01.” (page 6, line 179).” Multiple comparisons corrections were also indicated.
Reviewer 2 Report
This article is written in correct and understandable English. The results presentation is clear, the conclusion and the introduction adapted.
However I note many biases specially bias selection given by the voluntary participation of patients. This study is conducted on 2 units of child neurology center (level 3 of care) which leads to a potential bias on the severity level of neurological disorders.
- it would be really interesting to compare the questionnaires before and after the pandemic phase, or to know the parental stress level before the pandemic with regard to the chronic illness of their children. A control group with parents of children with non-neurological diseases would also have been appropriate in order to assess the real impact of lockdown on parental anxiety by differentiating neurodevelopmental disease from others.
- The classification of the different diseases should be reviewed, particularly in the epilepsy of unknown etiology group, which may include idiopathic epileptic diseases such as severe epileptic encephalopathy. Similarly, some children must fit into several boxes having both epilepsy and a neurodevelopmental disorder, for example. This leads to a real confounding bias for secondary analyses.
- In the results, the distribution of ages according to the pathology lacks interest and it would be more interesting to represent the distribution of the children who had a behavioral deterioration according to their disease.
-You have to put table 3 before Table 2.
Author Response
This article is written in correct and understandable English. The results presentation is clear, the conclusion and the introduction adapted.
We would acknowledge the reviewer for his positive judgement and for his valuable suggestions.
However I note many biases specially bias selection given by the voluntary participation of patients.
We totally agree with the reviewer and we had already cited this specific source of bias selection among the limitations of the study (page 8 line 258)
This study is conducted on 2 units of child neurology center (level 3 of care) which leads to a potential bias on the severity level of neurological disorders.
We totally agree with the reviewer, and we added this point among the limitations (page 9, lines 263-265)
it would be really interesting to compare the questionnaires before and after the pandemic phase, or to know the parental stress level before the pandemic with regard to the chronic illness of their children.
We better clarified among the limitations the lack of data about parental distress before and after the pandemic phase (page 9, lines 266-270). The collection of data after the pandemic phase was not originally planned because the date of the end of pandemic phase was not predictable when the original questionnaires were administered. The proposed comparison might be the aim of a future study (it cannot be realized without an amendment to the Ethic Committee).
A control group with parents of children with non-neurological diseases would also have been appropriate in order to assess the real impact of lockdown on parental anxiety by differentiating neurodevelopmental disease from others.
This control group, unfortunately, was not available. We added a specific statement about this limitation at the end of the discussion (page 9, lines 270-273)
- The classification of the different diseases should be reviewed, particularly in the epilepsy of unknown etiology group, which may include idiopathic epileptic diseases such as severe epileptic encephalopathy. Similarly, some children must fit into several boxes having both epilepsy and a neurodevelopmental disorder, for example. This leads to a real confounding bias for secondary analyses.
We were conscious that the proposed classification was not perfect and that it represented an additional source of bias selection, but it seemed a reasonable compromise to evaluate a so heterogeneous cohort.
We were not able to differentiate the patients according to the severity of the diseases because the anonymous self-reported questionnaire filled by the parents did not include information about severity. We had already explicated that participants were enrolled “without discriminations based on the severity of the diseases and the required frequency of controls”. (Page 4, lines 82-84). We added a further explicatory statement to highlight that “A predominant symptom, according to clinical judgement, was identified for the assignment of patients with comorbidities (e.g., epilepsy and neurodevelopmental disorders) to one of the abovementioned groups. Patients presenting with epilepsies with a known genetic etiology were assigned to the third group”. (Page 4, lines 103-107).
- In the results, the distribution of ages according to the pathology lacks interest and it would be more interesting to represent the distribution of the children who had a behavioral deterioration according to their disease.
We modified Fig. 1 according to reviewer’s suggestion and we included the distribution of patients with behavioral deterioration
-You have to put table 3 before Table 2.
We inverted the position of the two tables
Round 2
Reviewer 1 Report
No further comments.
Author Response
REPLY TO REVIEWER 2
This article is written in correct and understandable English. The results presentation is clear, the conclusion and the introduction adapted.
We would acknowledge the reviewer for his positive judgement and for his valuable suggestions.
However I note many biases specially bias selection given by the voluntary participation of patients.
We totally agree with the reviewer and we had already cited this specific source of bias selection among the limitations of the study (page 8 line 258)
This study is conducted on 2 units of child neurology center (level 3 of care) which leads to a potential bias on the severity level of neurological disorders.
We totally agree with the reviewer, and we added this point among the limitations (page 9, lines 263-265)
it would be really interesting to compare the questionnaires before and after the pandemic phase, or to know the parental stress level before the pandemic with regard to the chronic illness of their children.
We better clarified among the limitations the lack of data about parental distress before and after the pandemic phase (page 9, lines 266-270). The collection of data after the pandemic phase was not originally planned because the date of the end of pandemic phase was not predictable when the original questionnaires were administered. The proposed comparison might be the aim of a future study (it cannot be realized without an amendment to the Ethic Committee).
A control group with parents of children with non-neurological diseases would also have been appropriate in order to assess the real impact of lockdown on parental anxiety by differentiating neurodevelopmental disease from others.
This control group, unfortunately, was not available. We added a specific statement about this limitation at the end of the discussion (page 9, lines 270-273)
- The classification of the different diseases should be reviewed, particularly in the epilepsy of unknown etiology group, which may include idiopathic epileptic diseases such as severe epileptic encephalopathy. Similarly, some children must fit into several boxes having both epilepsy and a neurodevelopmental disorder, for example. This leads to a real confounding bias for secondary analyses.
We were conscious that the proposed classification was not perfect and that it represented an additional source of bias selection, but it seemed a reasonable compromise to evaluate a so heterogeneous cohort.
We were not able to differentiate the patients according to the severity of the diseases because the anonymous self-reported questionnaire filled by the parents did not include information about severity. We had already explicated that participants were enrolled “without discriminations based on the severity of the diseases and the required frequency of controls”. (Page 4, lines 82-84). We added a further explicatory statement to highlight that “A predominant symptom, according to clinical judgement, was identified for the assignment of patients with comorbidities (e.g., epilepsy and neurodevelopmental disorders) to one of the abovementioned groups. Patients presenting with epilepsies with a known genetic etiology were assigned to the third group”. (Page 4, lines 103-107).
- In the results, the distribution of ages according to the pathology lacks interest and it would be more interesting to represent the distribution of the children who had a behavioral deterioration according to their disease.
We modified Fig. 1 according to reviewer’s suggestion and we included the distribution of patients with behavioral deterioration
-You have to put table 3 before Table 2.
We inverted the position of the two tables